# Genotypic Variation in Anthocyanins, Phenolic Compounds, and Antioxidant Activity in Cob and Husk of Purple Field Corn

**Ponsawan Khamphasan [1], Khomsorn Lomthaisong [2], Bhornchai Harakotr [3], Danupol Ketthaisong [4,5], Marvin Paul Scott [6], Kamol Lertrat [5] and Bhalang Suriharn [1,5,\*]** 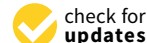

1   Department of Agronomy, Faculty of Agriculture, Khon Kaen University, Khon Kaen 40002, Thailand; ponsawan69@gmail.com
2   Department of Biochemistry, Faculty of Science, Khon Kaen University, Khon Kaen 40002, Thailand; kholom@kku.ac.th
3   Department of Agricultural Technology, Faculty of Science and Technology, Thammasat University, Phathum Thani 12120, Thailand; p.harakotr@gmail.com
4   Department of Horticulture, Faculty of Agriculture, Khon Kaen University, Khon Kaen 40002, Thailand; danupol63@gmail.com
5   Plant Breeding Research Center for Sustainable Agriculture, Khon Kaen 40002, Thailand; kamol9@gmail.com
6   USDA-ARS Corn Insects and Crop Genetics Research Unit, USDA-ARS, Ames, IA 50011, USA; Paul.Scott@ars.usda.gov
\*   Correspondence: bsuriharn@gmail.com; Tel.: +66-43-202-696

**Abstract:** Information on phytochemicals in the cob and husk of field corn is important for the use of corn waste in the production of value-added corn products. The objectives of this study were to evaluate the variation in monomeric anthocyanin content (MAC), total phenolic content (TPC), and antioxidant activity, as determined by 2,2-diphenyl-1-picrylhydrazyl (DPPH) free radical scavenging activity and Trolox equivalent antioxidant capacity (TEAC) in 53 purple field corn genotypes, and to study the correlations of these traits with color parameters. Fifty-three corn genotypes were planted in a randomized complete block design with three replications in two locations in the dry season of 2015/2016. The effects of genotype, location, and the interaction between genotype and location were significant for most characters. Genotypic variation contributed to a large portion of the total variance for all traits, accounting for 63.9–86.9%. Corn genotypes were classified into six groups based on MAC, TPC, and antioxidant activity determined by the DPPH and the TEAC methods. The highest MAC, TPC, and antioxidant activity were obtained in TB/KND//PF3 and TB/KND//PF8 for husk, and only TB/KND//PF8 for cob. They should be used as parental lines to develop corn varieties with high phytochemicals. Chroma (C\*) and hue (H°) of color parameters could potentially be used as an indirect selection criterion for improving MAC, TPC, and antioxidant activity in cob. The information is useful for the improvement of phytochemicals in cob and husk of field corn.

**Keywords:** *Zea mays* L.; maize; germplasm; diversity; phytochemicals; colorimeter; pH-differential

## 1. Introduction

Corn (*Zea mays* L.) is an important cereal, as it serves human needs for food, feed, and fuel [1,2]. Corn has many useful phytochemicals with health benefits such as carotenoids [3], flavonoids, phenolic compounds, and anthocyanins [4,5]. Purple corn is rich anthocyanin in cob [6,7], kernels [8–10], husk [6], and silk [11], Therefore, corn is beneficial to health beyond its role as an ordinary food, as

corn has useful phytochemicals and, currently, it can be used as a source of antioxidants for functional food products [6,7,12–14].

Antioxidants are attributed to scavenging reactive oxygen species (ROS) such as superoxide, singlet oxygen, peroxide, hydrogen peroxide, and hydroxyl radical which damage biomolecules in living cells [15]. Antioxidants from corn [14], blueberry [16], rice [17], and grape [18] can reduce the risk of developing chronic diseases such as cancer [14–18], cardiovascular disease [19,20], obesity [13], and diabetes [13,21], and they have remarkable potential health benefits [22].

The cob and husk are considered as waste in corn production. Although they can be used for animal feed and biofuels [23], the utilization of this agricultural waste is still limited. Understanding the content of beneficial compounds such as antioxidants in cob and husk may promote the utilization of this agricultural waste for production of value-added products. For corn, breeding for high anthocyanins in the cob and husk is a promising means to reduce waste in corn production and increase the value of corn production by-products.

Genetic diversity for the traits under improvement is a must for the success of breeding programs [24]. In 13 populations of temperate maize germplasm, variation in protein quality, nutrient, physical, and biochemical properties of starch and color have been reported [25]. Waxy corn germplasm also had high variation in anthocyanins and antioxidant activity in kernels [9], and had high variation in anthocyanin and antioxidant activity in cob and husk have been found in pod corn [6]. Anthocyanins are found in most parts of corn, including the cob [6,7], kernels [8–10], husk [6], and silk [11]. There is potential to convert normal field corn into high anthocyanin field corn using available germplasm. However, the available methods for screening of corn genotypes are still costly. The current methods of chemical analysis are not suitable for screening large numbers of plant populations or accessions.

Accurate and rapid methods are required for screening a large number of corn accessions for anthocyanin and antioxidants. A colorimetric system is a simple, rapid, cheap, and non-destructive alternative method for indirect selection for improving phytochemicals in a large population [26–28]. It was used as an indirect selection tool for improving the levels of anthocyanins and antioxidant activity in waxy corn kernel [9], carotenoid content in maize grain [28], and lycopene content in tomato [27]. However, information on the relationship between colorimetric data and antioxidant concentration and activity in husk and cob of field corn germplasm is still lacking. The research on phytochemicals of purple field corn genotypes in cob and husk is limited. Moreover, screening phytochemicals in the large populations used in breeding program is a laborious, time-consuming, and expensive process. The objectives of this study were to assess the variability of purple field corn for total anthocyanin content, total phenolic content, and antioxidant activities, and to study the correlations between these traits and color parameters. The information obtained in this study is useful to corn breeders who wish to improve antioxidant compounds and antioxidant capacity in the cob and husk of field corn.

## 2. Materials and Methods

### 2.1. Plant Materials

Near inbred lines (fourth or fifth generation of self-pollination) and check varieties of purple field corn, purple waxy corn, and normal field corn (Table 1) were evaluated in this study. Forty-seven lines (No. 1–47) are purple maize elite lines developed by Khon Kaen University, Thailand. Six commercial varieties (No. 48–53) were used as standard checks. KND Phitsanulok is purple field corn and KND KKU is purple waxy corn. These varieties have purple seeds, purple husks, and purple cobs. Oaxacan Green has green seeds, green husks, and brown cobs, and it is a commercial field corn variety from the United States. Fancy111 has purple seeds, purple-green husks and purple cobs, and it is a commercial purple waxy corn variety. Pacific339 and Pioneer4546 have orange seeds, green husks and white cobs, and they are commercial field corn varieties in Thailand.

**Table 1.** Fifty-three genotypes of purple field corn and waxy corn germplasm used in this study.

| No. | Varieties | Kernel Color | Cob Color | Husk Color | Country |
|---|---|---|---|---|---|
| 1 | AB/PF1 | Red-white | Purple | Purple | Thailand |
| 2 | PF/AB1 | Red-white | Purple | Purple | Thailand |
| 3 | PF/AB2 | Black | Purple | Purple | Thailand |
| 4 | PF/AB3 | Black | Purple | Purple | Thailand |
| 5 | PF/AB4 | white | Purple | Purple | Thailand |
| 6 | TB/KND//PF1 | Black | Purple | Purple | Thailand |
| 7 | TB/KND//PF2 | Black | Purple | Purple | Thailand |
| 8 | TB/KND//PF3 | Black | Purple | Purple | Thailand |
| 9 | TB/KND//PF4 | Black | Purple | Purple | Thailand |
| 10 | TB/KND//PF5 | Black | Purple | Purple | Thailand |
| 11 | TB/KND//PF6 | Black | Purple | Purple | Thailand |
| 12 | TB/KND//PF7 | Black | Purple | Purple | Thailand |
| 13 | TB/KND//PF8 | Black | Purple | Purple | Thailand |
| 14 | TB/KND//PF9 | Black | Purple | Purple | Thailand |
| 15 | TB/KND//PF10 | Black | Purple | Purple | Thailand |
| 16 | TB/KND//PF11 | Black | Purple | Purple | Thailand |
| 17 | TB/KND//PF12 | Black | Purple | Purple | Thailand |
| 18 | TB/KND//PF13 | Black | Purple | Purple | Thailand |
| 19 | TB/KND//PF14 | Black | Purple | Purple | Thailand |
| 20 | TB/KND//PF15 | Black | Purple | Purple | Thailand |
| 21 | TB/KND//PF16 | Black | Purple | Purple | Thailand |
| 22 | TB/KND//PF17 | Black | Purple | Purple | Thailand |
| 23 | TL/PF//KND10-1 | Black | Purple | Purple | Thailand |
| 24 | TL/PF//KND10-2 | Black | Purple | Purple | Thailand |
| 25 | TL/PF//KND10-3 | Black | Purple | Purple | Thailand |
| 26 | TL/PF//KND10-4 | white | Purple | Purple | Thailand |
| 27 | TL/PF//KND10-5 | Black | Purple | Purple | Thailand |
| 28 | TL/PF//KND10-6 | Black | Purple | Purple | Thailand |
| 29 | TL/PF//KND10-7 | Black | Purple | Purple | Thailand |
| 30 | TL/PF//KND10-8 | Black | Purple | Purple | Thailand |
| 31 | TL/PF//KND10-9 | Black | Purple | Purple | Thailand |
| 32 | TL/PF//KND10-10 | Black | Purple | Purple | Thailand |
| 33 | TL/PF//KND10-11 | Red | Purple | Purple | Thailand |
| 34 | TL/PF//KND10-12 | Yellow | Purple | Purple | Thailand |
| 35 | TL/PF//KND10-13 | Red | Purple | Purple | Thailand |
| 36 | TL/PF//KND10-14 | Red | Purple | Purple | Thailand |
| 37 | TL/PF//KND10-15 | white | Purple | Purple | Thailand |
| 38 | TL/PF//KND10-16 | Red | Purple | Purple | Thailand |
| 39 | TL/PF//KND10-17 | Red | Purple | Purple | Thailand |
| 40 | WSTS/PF//KND1 | Red | Purple | Purple | Thailand |
| 41 | WSTS/PF//KND2 | Red | Purple | Purple | Thailand |
| 42 | WSTS/PF//KND3 | Red-white | Purple | Purple | Thailand |
| 43 | WSTS/PF//KND4 | Red-white | Purple | Purple | Thailand |
| 44 | WSTS/PF//KND5 | White | Purple | Purple | Thailand |
| 45 | WSTS/PF//KND6 | Red-white | Purple | Purple | Thailand |
| 46 | WSTS/PF//KND7 | white | Purple | Purple | Thailand |
| 47 | WSTS/PF//KND8 | Red-white | Purple | Purple | Thailand |
| 48 | KND Phitsanulok | Black | Purple | Purple-Green | Thailand |
| 49 | KND KKU | Black | Purple | Purple-Green | Thailand |
| 50 | Oaxacan Green | Green | Brown | Green | United States |
| 51 | Fancy111 | Black | Purple | Purple-Green | Thailand |
| 52 | Pacific339 | Orange | white | Green | Thailand |
| 53 | Pioneer4546 | Orange | white | Green | Thailand |

*2.2. Field Experiment*

　　Fifty-three corn varieties were arranged in a randomized complete block design with three replications in the dry season (October 2015–March 2016) at two locations: the Research Station,

Khon Kaen University (16°28′11.24″ N 102°48′49.46″ E and altitude 190 m) and the farmer field in Uthai Thani Province (15°22′57.77″ N E 100° 4′42.54″ E and altitude 20 m), Thailand. The plot size had two 4 m long rows and a spacing of 0.8 m between rows and 0.25 m between plants within a row; recommended practices for the commercial production of corn were followed.

### 2.3. Sample Preparation and Extraction

Five ears from each accession in each replication were randomly harvested at physiological maturity (40 days after pollination) and oven-dried at 40 °C for 48 h. The anthocyanins in husk and cob were extracted according to the method described previously [29,30] with minor modifications. The harvested tissues from each replication were combined into husk and cob pools that were ground into powder separately, and approximately 2 g of the powdered samples were loaded into 100 mL flasks with containing 20 mL of 100% methanol. The flasks were shaken on a multi-stirrer (Model ST-1200EC, Diligent, Nonthaburi, Thailand) at 200 rpm for 1 h at room temperature. The samples were further filtered through Whatman # 1 filter paper.

After filtration, the retentates were loaded again into flasks with volume of 100 mL containing 20 mL of 100% methanol, shaken on a platform shaker for 1 h, and filtered through Whatman #1 filter paper. The filtrates were evaporated in a rotary evaporator (Rotavapor R-3, Buchi, Switzerland) to reduce the volume from 40 mL to 10 mL at 60 °C and stored at −20 °C in the dark.

### 2.4. Determination of Monomeric Anthocyanin Content (MAC)

Total monomeric anthocyanin content in each sample, which was separated into husk and cob, was estimated using the pH differential method [31]. A UV–vis spectrophotometer (GENESYS 10S, ThermoScientific, Waltham, MA, USA) was used to measure the absorbance at 510 and 700 nm in a cuvette with a 1 cm path length. Total monomeric anthocyanin concentration (MAC), total monomeric anthocyanin per husk dry weight of one ear (MAC/e), and total monomeric anthocyanin per cob dry weight in one ear (MAC/e) were expressed as mg of cyanidin-3-glucoside equivalents per 100 grams dry weight (mg CGE/100g DW) of samples, calculated by using the following equation;

$$\text{Anthocyanin pigment (cyanidin-3-glucoside equivalents, mg/L)} = (A \times MW \times DF \times 10^3)/(\varepsilon \times 1) \tag{1}$$

where $A$= (A510 nm − A700 nm)pH1.0 − (A510 nm − A700 nm)pH 4.5; MW (molecular weight) = 449.2 g/mol for cyanidin-3-glucoside (cyd-3-glu); $DF$ = dilution factor; 1 = pathlength in cm., $\varepsilon$ = 26,900 molar extinction coefficient, in L·mol$^{-1}$·cm$^{-1}$, for cyd-3-glu and $10^3$ = factor for conversion from g to mg.

### 2.5. Determination of Total Phenolic Content (TPC)

Phenolic content in each sample was determined according to Folin–Ciocalteau's phenol reagent (FC reagent) procedure with minor modification [32]. The reaction was prepared by mixing 0.5 mL methanol extract, 2.5 mL water, and 0.5 mL FC reagent, which was pre-diluted from 2 M to 1 M with distilled water. The mixture was set aside at room temperature for eight minutes and 1.5 mL $Na_2CO_3$ solution was added into the mixture. The mixture solution was allowed to stand for 120 min at room temperature. Then, the absorbance was read at 765 nm using a UV–visible spectrophotometer. A calibration curve was prepared using a standard solution of gallic acid (20, 40, 60, 80, and 100 μg/mL). The total phenolic content (TPC), total phenolic content per husk dry weight of one ear (TPC/e), and total phenolic content per cob dry weight in one ear (TPC/e) was expressed as mg gallic acid equivalents /100grams dry weight of samples (mg GAE/100g DW).

### 2.6. Determination of Antioxidant Assay

DPPH (2,2-diphenyl-1-picrylhydrazyl) free radical scavenging activity assay was determined by measuring the capacity of bleaching a black colored methanol solution of DPPH radicals as reported by [32]. Briefly, the reaction for each sample was prepared by mixing 4.5 mL methanolic solution of DPPH (0.065 mM) and 0.5 mL of solution extract or a standard solution. The reaction was conducted at room temperature for 30 min before the absorbance was recorded at 517 nm. The radical scavenging activity of the extracts was calculated as follows:

$$\text{scavenging rate (\%)} = (1 - ((A1 - As)/Ao)) \times 100 \qquad (2)$$

where $Ao$ is the absorbance of the control solution (0.5 mL extraction solvent in 4.5 mL of DPPH solution), $A1$ is the absorbance of the extracts in DPPH solution and $As$, which is used for error correction arising from unequal color of the sample solutions, is the absorbance of the extract solution without DPPH. The value was expressed percentage (%) of DPPH free radical scavenging activity assay.

Trolox equivalent antioxidant capacity assay (TEAC) for each sample was determined according to the method described [32] with minor modifications. Briefly, $ABTS^+$ radical cation was generated by a reaction of 7 mmol/L ABTS and 2.45 mmol/L potassium persulfate. The reaction mixture was allowed to stand in the dark at room temperature for 16–24 h before use; the mixture was used within 2 days. The $ABTS^+$ solution was diluted with methanol to an absorbance of $0.700 \pm 0.050$ at 734 nm). Fifty microliters of the diluted extract were mixed with 2.0 mL of diluted ABTS+ solution for 6 min at room temperature and the absorbance was immediately recorded at 734 nm. Trolox solution (100–1000 μM) was used as a reference standard. The value was expressed as micromoles Trolox equivalents (TE) per 100 grams of dry weight (μmol TE/100g DW).

### 2.7. Color Measurement

Color parameters were measured from five ears in each plot, which were separated into husk and cob, by HunterLab miniscan EZ colorimeter (Mod. MSEZ-4500L, Hunter Associates Laboratory Inc., Reston, VA, USA), and the colorimeter was calibrated prior to data collection with a HunterLab calibration standard white and black reflector plate. The color values for each sample of husk and cob were determined from five ears. Three husks from each ear were divided into nine pieces (top, middle, and bottom of the ear), whereas each cob was measured at three parts (top, middle, and bottom of the ear). The color was expressed as C* and H°. The chroma (C*) represented color intensity and hue (H°) expressed in degree range from 0° to 360° (0° = red, 90° = yellow, 180° = green and 270° = blue) [26].

### 2.8. Statistical Analysis

Individual analysis of variance (No. 1–47 lines) was performed for each character of each locations in husk and cob separately, and error variances were tested for homogeneity [33]. The statistical model is:

$$Y_{ijk} = m + B_i + L_j + G_k + LG_{jk} + e_{ij} + e_{ijk} \qquad (3)$$

where $Y_{ijk}$ was mean of genotypes $i$ in the location $j$ and block $k$, m was mean, $Bi$ was block effects, $L_j$ was locations effects, $G_k$ was genotypes effects, $LG_{jk}$ was interaction between locations and genotypes effects, $e_{ij}$ was locations error effects, and $e_{ijk}$ was pooled error effects. Least significant difference (LSD) was used to compare mean differences at 0.05 probability level. The correlation between color parameters (chroma and hue) vs. total anthocyanin content, total phenolic compounds, and antioxidant activities (the DPPH and the TEAC methods) was determined by Pearson's correlation analysis. Hierarchical agglomerative clustering was then performed for antioxidants and their activity using the Ward criterion with the JMPPro software (version 13.0, SAS institute Inc., Chicago, IL, USA).

## 3. Results and Discussion

### 3.1. Genotypic Variability

Differences in locations for both husk and cob were significant ($P \leq 0.01$) for monomeric anthocyanin content (MAC), monomeric anthocyanin content per ear (MAC/e), total phenolic content (TPC), total phenolic content per ear (TPC/e), 2,2-diphenyl-1-picrylhydrazyl radical scavenging ability (DPPH), Trolox equivalent antioxidant capacity (TEAC), and chroma (C*) except for hue (H°) in husk (Table 2). Difference in genotype and genotype by location interactions were significant ($P \leq 0.01$) for all traits.

**Table 2.** Mean squares and significance of effects for monomeric anthocyanin content (MAC), total phenolic content (TPC), and antioxidant activity determined by DPPH method and TEAC method and color parameters, in husk and cob of 1–47 genotypes evaluated in the dry season across Uthai Thani and Khon Kaen provinces.

| SOV | df | Antioxidants | | | | Antioxidant Capacity | | Color Parameters | |
|---|---|---|---|---|---|---|---|---|---|
| | | MAC | MAC/e | TPC | TPC/e | DPPH | TEAC | C* | H° |
| **Husk** | | | | | | | | | |
| Location (L) | 1 | 17,000,000 ** | 76,570 ** | 17,490,000 ** | 789,000,000 ** | 22,050 ** | 519,400,000 ** | 190.2 ** | 2.5 ns |
| | | (15.8) [a] | (8.3) | (10.4) | (4.5) | (16.0) | (8.0) | (23.2) | (0.0) |
| Rep/L (a) | 4 | 794 | 15 | 13,765 | 690,826 | 26 | 76,101 | 0.3 | 740.0 |
| | | (0.0) | (0.0) | (0.0) | (0.0) | (0.1) | (0.0) | (0.1) | (1.4) |
| Genotype (G) | 46 | 1,557,246 ** | 14,450 ** | 2,624,889 ** | 297,200,000 ** | 1,990 ** | 89,800,000 ** | 8.7 ** | 2044.6 ** |
| | | (66.7) | (72.4) | (71.6) | (77.7) | (66.2) | (63.9) | (48.8) | (45.2) |
| L × G | 46 | 396,767 ** | 3794 ** | 606,800 ** | 64,770,000 ** | 478 ** | 39,180,000** | 3.3 ** | 2256.4 ** |
| | | (17.0) | (19.0) | (16.5) | (17.0) | (15.9) | (27.8) | (18.5) | (49.8) |
| Error (b) | 184 | 3174 | 16 | 13,618 | 745,447 | 14 | 116,634 | 0.4 | 40.8 |
| | | (0.5) | (0.3) | (1.5) | (0.8) | (1.8) | (0.3) | (9.4) | (3.6) |
| C.V. (a) (%) | | 2.6 | 4.8 | 7.2 | 6.8 | 9.9 | 2.3 | 4.2 | 9.4 |
| C.V. (b) (%) | | 5.3 | 5.0 | 7.2 | 7.0 | 7.2 | 2.8 | 5.3 | 2.2 |
| **Cob** | | | | | | | | | |
| Location (L) | 1 | 2,913,055 ** | 8091 ** | 3,619,754 ** | 46,040,000 ** | 1725 ** | 198,400,000 ** | 87.3 ** | 382.9 ** |
| | | (2.6) | (1.0) | (2.4) | (0.4) | (1.7) | (3.0) | (4.7) | (1.4) |
| Rep/L (a) | 4 | 1666 | 4.4 | 2758 | 127,264 | 5 | 257,194 | 0.4 | 1.4 |
| | | (0.0) | (0.0) | (0.0) | (0.0) | (0.0) | (0.0) | (0.1) | (0.0) |
| Genotype (G) | 46 | 2,003,180 ** | 14,920 ** | 2,855,735 ** | 244,100,000 ** | 1873 ** | 121,900,000 ** | 30.1 ** | 430.9 ** |
| | | (83.2) | (84.9) | (85.3) | (86.9) | (86.9) | (86.1) | (74.6) | (70.0) |
| L × G | 46 | 340,432 ** | 2449 ** | 385,483 ** | 33,370,000 ** | 231 ** | 14,540,000 ** | 6.2 ** | 134.4 ** |
| | | (14.1) | (13.9) | (11.5) | (11.9) | (10.7) | (10.3) | (15.5) | (21.9) |
| Error (b) | 184 | 597 | 9 | 6835 | 579,652 | 4 | 217,930 | 0.5 | 10.3 |
| | | (0.1) | (0.2) | (0.8) | (0.8) | (0.7) | (0.6) | (5.1) | (6.7) |
| C.V. (a) (%) | | 4.1 | 2.8 | 3.8 | 3.3 | 5.5 | 5.4 | 6.5 | 0.4 |
| C.V. (b) (%) | | 2.5 | 4.0 | 6.0 | 7.1 | 5.0 | 5.0 | 7.0 | 1.1 |

Genotypes 1–47 were calculated analysis of variance without check varieties. SOV source of variation, *df* degree of freedom, *MAC* monomeric anthocyanin content *MAC/e* monomeric anthocyanin content per ear, *TPC* total phenolic content, *TPC/e* total phenolic content per ear, DPPH 2,2-diphenyl-1-picrylhydrazyl radical scavenging ability, *TEAC* Trolox equivalent antioxidant activity, C* Chroma, and H° Hue angle. ns, ** non-significant and Significant at $P \leq 0.05$ and 0.01, respectively. [a] number within the parentheses is percentage of sum squares to total sum of squares.

In husk, location showed low to moderate contribution to total variations of MAC, MAC/e TPC, TPC/e, and antioxidant activity (4.5%–15.8%) and color parameters (0.0%–23.2%). Genotype contributed to a large portion of total variations in TPC/e (77.7%), MAC/e (72.4%), TPC (71.6%), MAC (66.7%), DPPH (66.2%), TEAC (63.9%), C* (48.8%), and H° (45.2%). However, the contributions of interaction between genotype and location to total variations were low to moderate for color parameter (18.5%–49.8%) and MAC, MAC/e TPC, TPC/e and antioxidant activity (15.9%–27.8%).

In cob, location made small contributions to total variance of MAC, MAC/e TPC, TPC/e, and antioxidant activity (0.4%–3.0%), as well as color parameters (1.4%–4.7%). Genotype contributed to a large portion of total variance in TPC/e (86.9%), DPPH (86.9%), TEAC (86.1%), TPC (85.3%), MAC/e (84.9%), MAC (83.2%), C* (74.6%), and H° (70.0%). However, most interactions between genotype and location were low to moderate for color parameter (15.5%–21.9%) and MAC, MAC/e TPC, TPC/e, and antioxidant activity (10.3%–14.1%).

In this study, although locations were significantly different for most traits except for hue, location contributed to the smallest portions of total variations for these traits, and the contribution of genotype by location interaction was also low compared to the highest contribution of genotype. Low contribution of genotype by location interaction indicated that the genotypes performed consistently across locations. In other words, the genotypes that performed poorly at one location also performed poorly in other locations and vice versa. However, the presence of significant interactions of genotypes by location also indicated the inconsistent performance of some genotypes across locations. Therefore, genotypes require extensive evaluation in several locations.

Direct comparison with other studies for the effects of location is not possible, as the authors could not find related reports in the literature. However, season also had a small effect on MAC, TPC, DPPH, and TEAC in waxy corn kernel [9], and environment had small effects on phytochemicals related to antioxidants in cereals [34]. Low genotype by location interactions in this study were in agreement with those in the previous studies, confirming the low effects of environment on the variations in these traits. Although the effect of genotype by location interactions were low, the presence of significant effects might be due to noise from the selection programs, and hinder the progress of selection of field corn breeding for increasing monomeric anthocyanin content, total phenolic content, and antioxidant activity determined by the DPPH and the TEAC methods in both husk and cob.

Genotypes were significantly different for MAC, MAC/e, TPC, TPC/e, DPPH, and TEAC in both husk and cob, and genotype contributed the largest portions of total variance for these traits, ranging from 45.2% to 77.7% in husk and 70.0% to 86.9% in cob. In previous studies in corn kernels, genotype also had higher contribution to total variance for these traits than did environment and genotype by environment interactions [9,35,36]. The results in this study agreed with those in the previous studies, although direct comparison of the results for corn cob and husk is not possible. High variation among genotypes for these traits suggested it is possible to improve these traits through conventional corn breeding.

The contributions of genotype by location interaction to total variance in MAC, MAC/e, TPC, TPC/e, DPPH, and TEAC were low to moderate, from 15.9% to 49.8% in husk and 10.3% to 21.9% in cob. The previous studies reported that the genotype by environment interaction had significant effects on anthocyanins and antioxidant activity in waxy corn cob [8], physicochemical components in white land lace maize [37], lutein, α-carotene, β-carotene, and pro-vitamin A content in tropical adapted maize [38] and grain yield in maize [39] and in double haploid-hybrid maize [40]. However, analysis of variance in each location in this study indicated that only few genotypes in group A performed consistently across locations for anthocyanin content (Data not shown). The variations caused by genotype and environment interactions are important for selection of superior genotypes, and evaluation in multiple locations is still required.

*3.2. Cluster Analysis*

A dendrogram based on MAC, TPC, DPPH and TEAC classified 1–47 genotypes and 6 check varieties into six distinct groups (Figure 1). Group A consisted of 11 genotypes (TB/KND//PF3, TB/KND//PF7, TB/KND//PF8, TB/KND//PF9, TB/KND//PF10, TB/KND//PF11, TB/KND//PF12, TB/KND//PF14, TB/KND//PF15, TB/KND//PF16, and TL/PF//KND10-11) (Figure 2b). Most of them had the highest values of MAC, TPC, DPPH, and TEAC in husk and cob (Table 3). TB/KND//PF3 and TB/KND//PF8 were the best genotypes in this group. These genotypes should be used as parental lines to cross with high yielding varieties to create base populations for further population improvement using appropriate selection strategies.

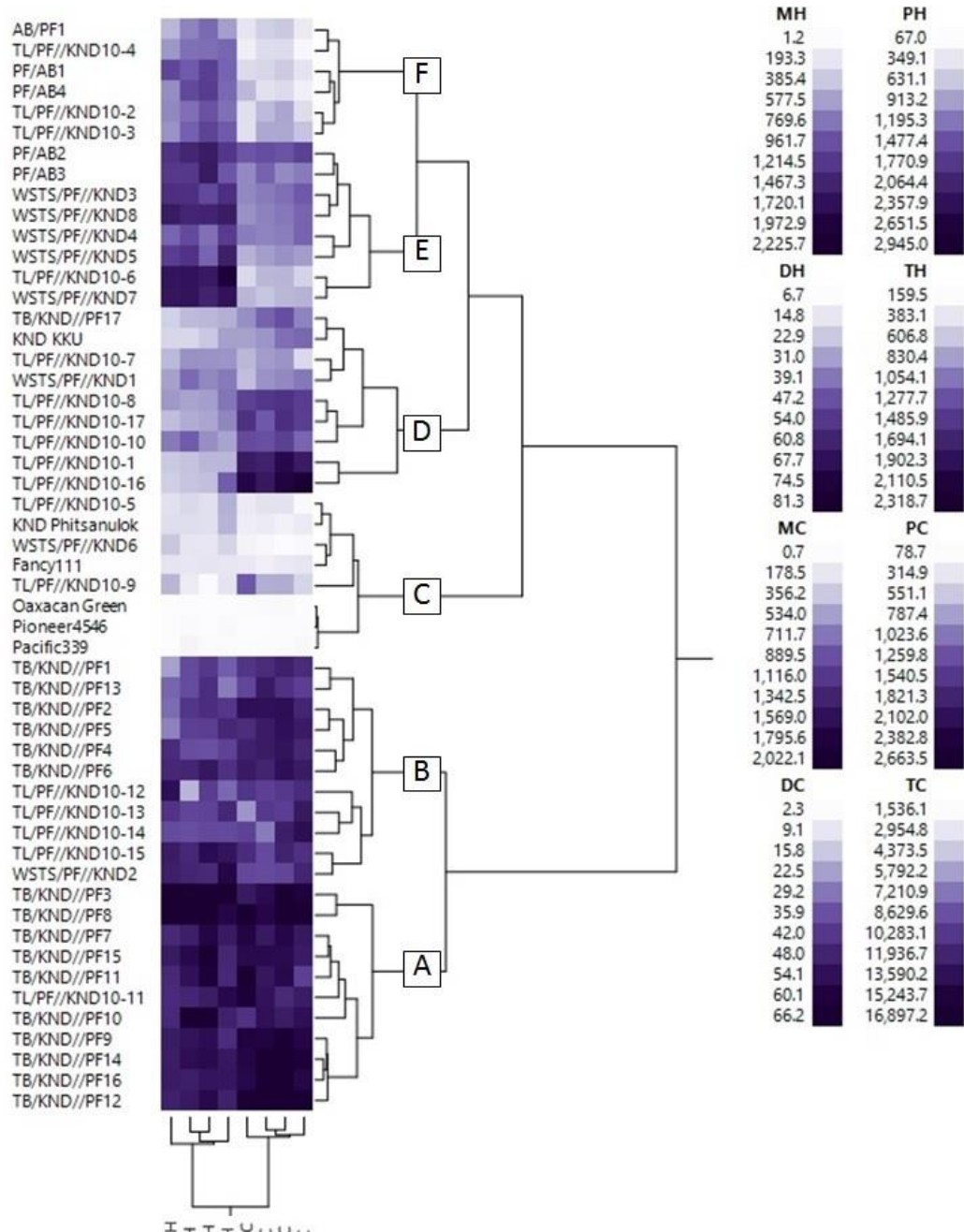

**Figure 1.** Dendrogram of genetic relationships among 53 genotypes. Two ways Ward's cluster analysis based on antioxidants and antioxidant activities traits. Six main clusters (A to F) were formed, *MH* monomeric anthocyanin content on husk, *MC* monomeric anthocyanin content on cob, *PH* total phenolic content on husk, *PC* total phenolic content on cob, *DH* 2,2-diphenyl-1-picrylhydrazyl radical scavenging ability on husk, *DC* 2,2-diphenyl-1-picrylhydrazyl radical scavenging ability in cob, *TH* Trolox equivalent antioxidant capacity in husk, *TC* Trolox equivalent antioxidant capacity on cob.

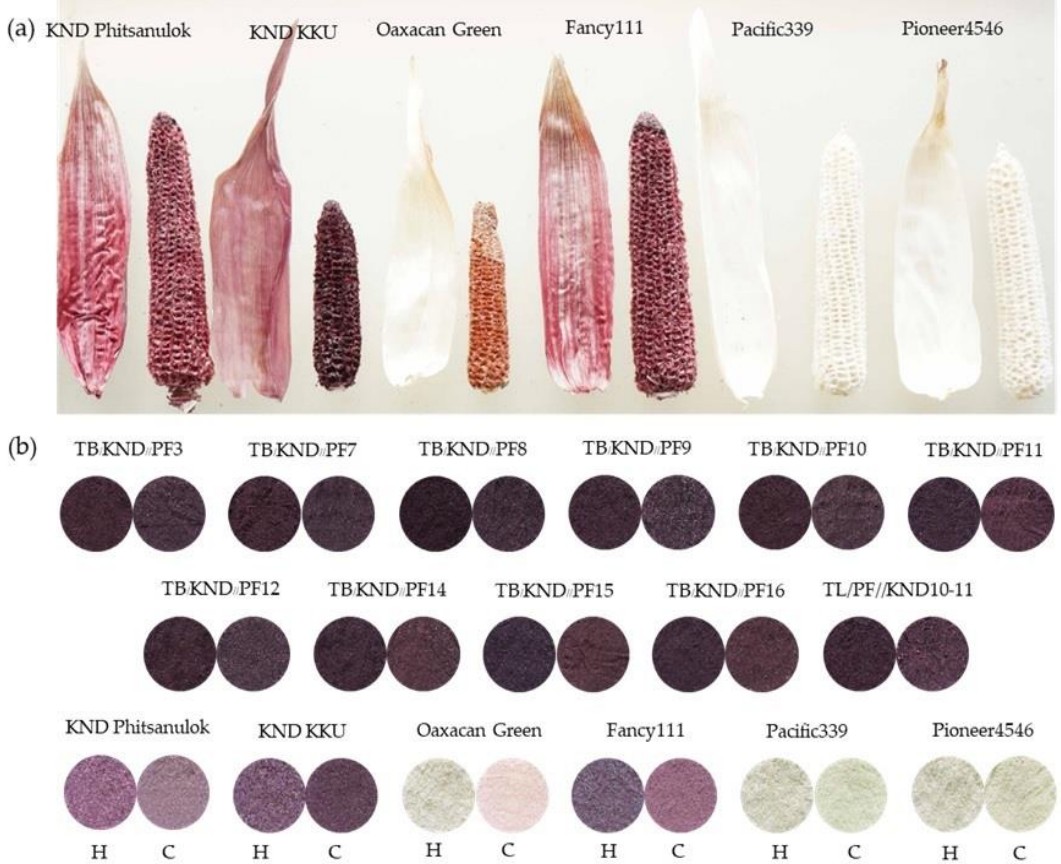

**Figure 2.** Husk (H) and cob (C) of (**a**) check varieties and (**b**) powdered samples of group A and check varieties.

Group B consisted of 11 genotypes (TB/KND//PF1, TB/KND//PF2, TB/KND//PF4, TB/KND//PF5, TB/KND//PF6, TB/KND//PF13, TL/PF//KND10-12, TL/PF//KND10-13, TL/PF//KND10-14, TL/PF//KND10-15, and WSTS/PF//KND2). This group had medium values of MAC, TPC DPPH, and TEAC in husk and cob.

Group C consisted of 8 genotypes (TL/PF//KND10-5, TL/PF//KND10-9, WSTS/PF//KND6, KND Phitsanulok, Oaxacan Green, Fancy11, Pacific339 and Pionerr4546). This group showed the lowest values of MAC, TPC, DPPH, and TEAC in husk and cob. Group D consisted of 9 genotypes (TB/KND//PF17, TL/PF//KND10-1, TL/PF//KND10-7, TL/PF//KND10-8, TL/PF//KND10-10, TL/PF//KND10-16, TL/PF//KND10-17, WSTS/PF//KND-1, and KND KKU). Most of them had medium to low values of MAC, TPC DPPH, and TEAC in husk, and medium to low MAC, TPC DPPH, and TEAC in cob.

Group E consisted of 8 genotypes PF/AB2, PF/AB3, TL/PF//KND10-6, WSTS/PF//KND3, WSTS/PF//KND4, WSTS/PF//KND5, WSTS/PF//KND7 and WSTS/PF//KND8). Most genotypes in this group had medium to high values of MAC, TPC, DPPH, and TEAC in husk, and medium values of MAC, TPC, DPPH, and TEAC in cob.

Group F consisted of 6 genotypes (AB/PF1, PF/AB1, PF/AB4, TL/PF//KND10-2, TL/PF//KND10-3, and TL/PF//KND10-4). This group had medium values of MAC, TPC, DPPH, and TEAC in husk and low values of MAC, TPC, DPPH, and TEAC in cob.

The effects of genotype by location interaction on MAC, MAC/e, TPC, TPC/e, DPPH, and TEAC were low to moderate in 6 group (A–F). In contrast, in earlier research on anthocyanin in waxy corn kernels, genotype by season interaction was shown to have a low effect [9]. However, small but significant interactions between genotype and environment might hinder the progress of selection programs, and multiple-location trials are necessary to identify the superior genotypes for these traits.

**Table 3.** Means for monomeric anthocyanin content, total phenolic content, and antioxidant activity determined by *DPPH* 2,2-diphenyl-1-picrylhydrazyl radical scavenging ability and *TEAC* Trolox equivalent antioxidant capacity in husk of 11 genotypes (Group A) and 6 check varieties averaged over two locations.

| Husk | | | | | | |
|---|---|---|---|---|---|---|
| Group A [1] | MAC | MAC/e | TPC | TPC/e | DPPH | TEAC |
| TB/KND//PF3 | 2093.6 | 203.9 | 2945.0 | 28,526.0 | 81.3 | 20,085.0 |
| TB/KND//PF7 | 1389.4 | 108.2 | 2137.3 | 16,080.0 | 72.8 | 15,182.0 |
| TB/KND//PF8 | 2225.8 | 180.6 | 2797.2 | 22,492.0 | 78.2 | 17,335.0 |
| TB/KND//PF9 | 1561.4 | 148.3 | 2308.1 | 22,042.0 | 69.5 | 14,307.0 |
| TB/KND//PF10 | 1387.1 | 139.6 | 2804.4 | 28,118.0 | 77.4 | 15,092.0 |
| TB/KND//PF11 | 1384.5 | 110.3 | 2326.4 | 18,860.0 | 75.5 | 14,349.0 |
| TB/KND//PF12 | 1483.6 | 112.0 | 2210.2 | 16,609.0 | 70.7 | 14,467.0 |
| TB/KND//PF14 | 1539.9 | 137.5 | 2398.1 | 21,256.0 | 70.3 | 15,328.0 |
| TB/KND//PF15 | 1566.3 | 96.1 | 2484.7 | 15,153.0 | 76.8 | 14,293.0 |
| TB/KND//PF16 | 1564.0 | 103.0 | 2132.5 | 14,128.0 | 66.0 | 15,147.0 |
| TL/PF//KND10-11 | 1361.8 | 42.7 | 2024.7 | 6323.0 | 63.2 | 17,064.0 |
| LSD | 64.0 | 4.6 | 132.9 | 983.5 | 4.2 | 389.0 |
| **Check Varieties** | | | | | | |
| KND Phitsanulok | 274.4 | 50.0 | 461.7 | 8403.0 | 17.6 | 6349.0 |
| KND KKU | 323.7 | 60.6 | 534.8 | 10,039 | 23.2 | 7619.0 |
| Oaxacan green | 3.3 | 0.4 | 67.3 | 847.0 | 6.8 | 1272.0 |
| Fancy 111 | 214.6 | 32.1 | 349.6 | 5227.0 | 15.6 | 3279.0 |
| Pacific339 | 1.1 | 0.3 | 167.5 | 4756.0 | 6.9 | 1453.0 |
| Pioneer4546 | 1.8 | 0.4 | 92.0 | 2224.0 | 6.7 | 1550.0 |

Least significant difference (LSD) was used to compare mean differences at $P \leq 0.05$ level of 1–47 genotypes without check varieties. [1] The corn genotypes were classified into group A and check varieties based on *MAC* monomeric anthocyanin content (mg CGE/g 100 DW) *MAC/e* monomeric anthocyanin content per ear (mg CGE/100g DW), *TPC* total phenolic content (mg GAE/100g DW), *TPC/e* total phenolic content per ear (mg GAE/100g DW), *DPPH* 2,2-diphenyl-1-picrylhydrazyl radical scavenging ability (%), *TEAC* Trolox equivalent antioxidant activity (µmol TE/100g DW).

*3.3. Monomeric Anthocyanin Content*

In husk, as the error variances for all characters were homogenous, the data of two locations were averaged, and combined data of two locations for each genotype were reported. The dendrogram was able to classify the high value group (group A), and they were significantly different from the low value group (group C) and higher than check varieties (Table 3). The highest MAC in husk was observed in TB/KND//PF8 (2225.8 mg CGE/100g DW) and TB/KND//PF3 (2093.6 mg CGE/100g DW), respectively. TB/KND//PF8 and TB/KND//PF3 were also the highest genotypes for MAC/e in husk.

In cob, the dendrogram classified the high group and the low group, and these groups were significantly different for MAC in cob. In the high value group (group A), the data averaged from two locations showed that TB/KND//PF8 had the highest monomeric anthocyanin contents in cob (2022.1 mg CGE/100g DW), and it was significantly different from low value group (group C) and higher than check varieties, whereas TB/KND//PF7 had the highest MAC/e (204.9 mg CGE/100g DW) (Table 4).

**Table 4.** Means for monomeric anthocyanin content, total phenolic content and antioxidant activity determined by *DPPH* 2,2-diphenyl-1-picrylhydrazyl radical scavenging ability and *TEAC* Trolox equivalent antioxidant capacity in cob of 11 genotypes (Group A) and 6 check varieties averaged over two locations.

| **Cob** | | | | | | |
|---|---|---|---|---|---|---|
| Group A [1] | MAC | MAC/e | TPC | TPC/e | DPPH | TEAC |
| TB/KND//PF3 | 1455.4 | 129.0 | 2179.9 | 19,390.0 | 62.2 | 15,224.0 |
| TB/KND//PF7 | 1749.1 | 204.9 | 1955.2 | 22,852.0 | 57.9 | 12,821.0 |
| TB/KND//PF8 | 2022.1 | 140.4 | 2277.4 | 15,594.0 | 66.2 | 16,849.0 |
| TB/KND//PF9 | 1787.4 | 157.8 | 2334.5 | 20,807.0 | 61.8 | 14,647.0 |
| TB/KND//PF10 | 1176.5 | 118.4 | 2090.4 | 20,862.0 | 50.1 | 13,619.0 |
| TB/KND//PF11 | 1873.6 | 198.9 | 2002.1 | 21,383.0 | 56.6 | 9675.0 |
| TB/KND//PF12 | 1854.5 | 157.9 | 2663.7 | 23,108.0 | 64.9 | 16,897.0 |
| TB/KND//PF14 | 1733.3 | 136.4 | 2545.6 | 20,410.0 | 65.4 | 15,460.0 |
| TB/KND//PF15 | 1654.4 | 89.4 | 2211.5 | 12,288.0 | 55.8 | 12,613.0 |
| TB/KND//PF16 | 1677.8 | 64.9 | 2551.8 | 9897.0 | 64.1 | 14,571.0 |
| TL/PF//KND10-11 | 1842.3 | 86.3 | 2066.2 | 9831.0 | 46.0 | 12,496.0 |
| LSD | 27.8 | 3.4 | 94.2 | 867.2 | 2.2 | 531.8 |
| **Check Varieties** | | | | | | |
| KND Phitsanulok | 122.7 | 22.3 | 274.1 | 5041.0 | 5.4 | 2710.0 |
| KND KKU | 530.1 | 92.8 | 849.5 | 14,868 | 30.3 | 7696.0 |
| Oaxacan green | 7.4 | 1.1 | 96.9 | 1401.0 | 3.3 | 1536.0 |
| Fancy 111 | 158.4 | 29.2 | 209.0 | 3575.0 | 7.8 | 2960.0 |
| Pacific339 | 0.7 | 0.2 | 93.6 | 2610.0 | 2.9 | 1738.0 |
| Pioneer4546 | 0.9 | 0.2 | 78.6 | 1948.0 | 2.7 | 1606.0 |

Least significant difference (LSD) was used to compare mean differences at $P \leq 0.05$ level of 1–47 genotypes without check varieties. [1] The corn genotypes were classified into group A and check varieties based on *MAC* monomeric anthocyanin content (mg CGE/g 100 DW) *MAC/e* monomeric anthocyanin content per ear (mg CGE/100g DW), *TPC* total phenolic content (mg GAE/100g DW), *TPC/e* total phenolic content per ear (mg GAE/100g DW), *DPPH* 2,2-diphenyl-1-picrylhydrazyl radical scavenging ability (%), *TEAC* Trolox equivalent antioxidant activity (μmol TE/100g DW).

In this study, husk had higher MAC (2225.8mg/100g DW) than did cob (2022.1 mg/100g DW). The ranges of MAC in cob and husk of field corn in this study was higher than that in cob of purple field corn [7,29,41] and purple waxy corn [42], but lower than in husk of pod corn [6]. The range of MAC in corn cob and husk in this study was much higher than in purple carrot (44–57 mg/100g DW), red cabbage (198 mg/100g DW), purple cauliflower Graffiti (201 mg/100g DW), purple potato (48–97 mg/100g DW) [43], black rice (327 mg/100g DW) [44], and black rice bran (256 mg/100g DW) [18].

The ranges of anthocyanins in corn varied depending on plant parts such as kernel (106 to 680 mg/100g DW) [9,10,45], husk (18,900 mg/100g DW) [6], cob (34 to 1333 mg/100g DW) [29,30,42], and silk (2.3–6.3 mg/100g DW) [10,11]. It is interesting to note here that husk and cob had higher anthocyanin than did other parts of corn. Therefore, breeding for improving the levels of anthocyanin in husk and cob will increase the value of waste in corn production.

*3.4. Total Phenolic Content*

In husk, on the averages of two locations, TB/KND//PF3 (2945.0 mg GAE/100g DW), TB/KND//PF10 (2804.4 mg GAE/100g DW) and TB/KND//PF8 (2797.2 mg GAE/100g DW) had the highest TPC in husk, and the high value group (group A) were significantly higher than the low value group (group C) and check varieties. These genotypes were also highest for TPC/e in husk (Table 3).

In cob, on the averages of two locations, the highest TPC in cob were observed in the high value group (group A),the data averaged from two locations showed that TB/KND//PF12 had the highest TPC in cob (2663.7 mg GAE/100g DW); it was significantly different from low value group (group C) and check varieties, whereas the highest TPC/e were observed in TB/KND//PF12

(23,108.0 mg GAE/100g DW) and TB/KND//PF7 (22,852.0 mg GAE/100g DW) (Table 4). The high groups (group A) for TPC and TPC/e were significantly higher than the low value group (group C).

## 3.5. Antioxidant Activity

In husk, on the averages of two locations, the highest antioxidant activity values determined by DPPH method were observed in TB/KND//PF3 (81.3%), TB/KND//PF8 (78.2%), and TB/KND//PF 10 (77.4%) (Table 3). The high value group (group A) had the antioxidant activities values ranging from 63.2% to 81.3%, which were significantly higher than the low value group (group C) and check varieties.

On the averages of two locations, the highest values of antioxidant activity determined by TEAC method were observed in TB/KND//PF3 (20,085.0 μmol TE/100g DW) and TB/KND//PF 10 (15,092.0 μmol TE/100g DW) (Table 3). The antioxidant activities in the high value group (group A) were significantly higher than in the low value group (group C) and check varieties.

Antioxidant activity in cob was determined by the DPPH and the TEAC methods. TB/KND//PF8 (66.2%), TB/KND//PF14 (65.4%), and TB/KND//PF12 (64.9%) were the highest genotypes for DPPH, whereas TB/KND//PF12 (16,897.0 μmol TE/100g DW) and TB/KND//PF8 (16,849.0 μmol TE/100g DW) were the highest genotypes for TEAC (Table 4).

## 3.6. Correlation

### 3.6.1. Color Parameters vs. Antioxidant Content Relationship

It is possible to use color parameters (C* and H°) as a selection tool for phytochemical concentration and antioxidant activity if the correlations are high enough. Monomeric anthocyanin content in husk, phenolic content in husk, antioxidant activity determined by the DPPH and the TEAC methods in husk, monomeric anthocyanin content in husk/ear, and phenolic content in husk/ear were negatively and significantly correlated with chroma in husk ranging from −0.31to 0.54 and hue in husk ranging from −0.12 to 0.19, whereas monomeric anthocyanin content in cob, phenolic content in cob, antioxidant activity determined by the DPPH and the TEAC methods in cob, monomeric anthocyanin content in cob/ear, and phenolic in cob/ear were not correlated with chroma in husk and hue in husk ranging from −0.05 to 0.1 (Table 5).

**Table 5.** Pearson correlation coefficients between color parameters, monomeric anthocyanin content, total phenolic content, and antioxidant activities of 1–47 genotypes in husk and cob.

| | **Color Parameters** | | | |
| --- | --- | --- | --- | --- |
| | **C* in Husk** | **C* in Cob** | **H° in Husk** | **H° in Cob** |
| MAC in husk | −0.54 ** | −0.42 ** | −0.12 * | −0.33 ** |
| MAC in cob | −0.02 ns | −0.69 ** | 0.10 ns | −0.55 ** |
| TPC in husk | −0.45 ** | −0.47 ** | −0.18** | −0.31 ** |
| TPC in cob | 0.00 ns | −0.73 ** | −0.05 ns | −0.60 ** |
| DPPH in husk | −0.45 ** | −0.51 ** | −0.14 * | −0.36 ** |
| DPPH in cob | 0.01 ns | −0.74 ** | −0.05 ns | −0.65 ** |
| TEAC in husk | −0.48 ** | −0.37 ** | −0.16 ** | −0.32 ** |
| TEAC in cob | 0.00 ns | −0.68 ** | −0.06 ns | −0.60 ** |
| MAC in husk/e | −0.39 ** | −0.39 ** | −0.17 ** | −0.30 ** |
| MAC in cob/e | 0.03 ns | −0.58 ** | −0.06 ns | −0.48 ** |
| TPC in husk/e | −0.31 ** | −0.40 ** | −0.19 ** | −0.28 ** |
| TPC in cob/e | 0.02 ns | −0.59 ** | −0.11 ns | −0.52 ** |

The Pearson's correlation coefficients were calculated to determine the relationships among character of 1–47 genotypes without check varieties. C* chroma, H° hue angle, *MAC* monomeric anthocyanin content, *MAC/ear* monomeric anthocyanin content per ear, *TPC* total phenolic content, *TPC/ear* total phenolic content per ear, *DPPH* 2,2-diphenyl-1-picrylhydrazyl radical scavenging ability, *TEAC* Trolox equivalent antioxidant capacity. ns, *, ** non-significant and Significant at $P \leq 0.05$ and $0.01$, respectively.

Color parameters have been used to measure chromaticity values of food in industry and samples of plants in plant breeding for high β-carotene and lycopene in tomato [27] and high carotenoid in maize [28]. In this study, chroma in cob had moderate correlations with monomeric anthocyanin content, total phenolic content and antioxidant activity determined by the DPPH and the TEAC methods in cob, ranging from $-0.68$ to $-0.74$. Hue in cob had moderate correlations with monomeric anthocyanin content, total phenolic content, and antioxidant activity determined by the DPPH and the TEAC methods in cob, ranging from $-0.55$ to $-0.65$ (Table 5). In a previous study, the color parameter chroma and hue was closely related to monomeric anthocyanin content, total phenolic content, and antioxidant activity determined by the DPPH and the TEAC methods in corn kernels [9]. The results in this study agreed with those in previous studies. Therefore, chroma and hue could be used as indirect selection for monomeric anthocyanin content, total phenolic content, and antioxidant activity determined by the DPPH and the TEAC methods in corn cob. In contrast to the color parameter, the correlations between chroma and hue with monomeric anthocyanin content, total phenolic content, and antioxidant activity determined by the DPPH and the TEAC methods in corn husk were low. Therefore, chroma and hue should not be used as indirect criteria for selection of these traits in husk.

### 3.6.2. Antioxidant Content vs. Antioxidant Activity Relationship

All phytochemicals and antioxidant activity (both in husk and cob) were negatively and significantly correlated with chroma in cob ranging from $-0.39$ to $-0.74$ and hue in cob from $-0.28$ to $-0.65$. Monomeric anthocyanin content in husk and cob was significantly and positively correlated with Monomeric anthocyanin content per ear, total phenolic content, total phenolic content per ear, and antioxidant activity determined by the DPPH and the TEAC methods. Similar results were also reported in blueberry, cranberry, blackberry [46], and purple waxy corn kernel [8–10]. The results suggested that selection for high anthocyanins will result in the increase in total phenolic content and antioxidant activity.

### 4. Conclusions

The corn genotypes clusters were classified into six groups based on phytochemicals and antioxidant activity. TB/KND//PF3 and TB/KND//PF8 had the highest MAC, TPC, and antioxidant activity in husk. TB/KND//PF8 had the highest MAC, TPC, and antioxidant activity in cob. Chroma and hue in cob were closely correlated with MAC, TPC, and antioxidant activity, and this parameter may be useful as a selection criterion for these traits. The information obtained in this study is important for reducing waste in corn production.

**Author Contributions:** Conceptualization, P.K., B.H., D.K., Ka.L. and B.S.; Formal analysis, P.K., Kh.L., M.P.S. and B.S.; Methodology, P.K., Kh.L., B.H., Ka.L. and B.S.; Writing-original draft, P.K. and B.S.; Writing—review & editing, D.K, Kh.L., B.H., Ka.L. and M.P.S.

**Funding:** The Thailand Research Fund through the Royal Golden Jubilee Ph.D. Program (Grant No PHD/0014/2557).

**Acknowledgments:** The study was supported by the Thailand Research Fund through the Royal Golden Jubilee Ph.D. Program (Grant No PHD/0014/2557). The authors would like to thank the National Science and Technology Development Agency through the National Center for Genetic Engineering and Biotechnology, Bangkok, Thailand and the Plant Breeding Research Center for Sustainable Agriculture, Faculty of Agriculture, Khon Kaen University, Thailand. The acknowledgment was extended to the Thailand Research Fund (Project code: IRG5780003), Khon Kaen University and the Faculty of Agriculture KKU for providing financial support for manuscript preparation activities.

**Conflicts of Interest:** The authors declare no conflict of interest. The founding sponsors had no role in the design of the study; in the collection, analyses, or interpretation of data; in the writing of the manuscript, and in the decision to publish the results.

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
