# Peer review of "Genotypic Variation in Anthocyanins, Phenolic Compounds, and Antioxidant Activity in Cob and Husk of Purple Field Corn"

_agronomy, doi:10.3390/agronomy8110271_

Round 1

Reviewer 1 Report

The manuscript, agronomy-381979 dealt with the genetic variation of several phenolic compounds and antioxidant activities in cob and husk (not in kernels) of purple maize genotypes, and tried to search useful indicators of antioxidant contents. The contents of this manuscript might be useful in further breeding of maize, and I think this manuscript could basically be accepted as a regular paper in Agronomy. However, this manuscript involved many ambiguous parts to be revised. Particularly, the paragraph from line 425-433 should be revised completely, or be deleted. Some data should be missing. Also, I feel redundancy in the context of results and discussion. Please revise the manuscript more and more concisely and compactly. I concluded that this manuscript should be re-submitted after major revision. The individual problems are as follows:

Line 31: highest MAC, TPC and antioxidant activities was obtained in TB/KND//PF3 and TB/KND//PF8 for husk and only TB/KND//PF8 for cob.

Line 33: What are chroma and hue? “Chroma (C*) and hue (H°) of color parameter could potentially … “ (for example)

Line 47: Antioxidants from corn, for example, can reduce the … (for example, I cannot understand “These”.)

Line 57: Does “field” corn mean flint, dent or others?

Line 63: What are the current methods? Describe briefly.

Line 66: anthcyanins

Line 89: I think the format of Table 1 is not correct: draw a line between the first (no.) and second (1) rows.

Line 92: at two locations: at the …

Line 95: two rows of 4 m long

Line 96: within a row

Line 97-: Please show the production company for main chemicals including standards used, like Line 111.

Line 98: from each accession in each replication

Line 102 and others: ml => mL

Line 117: Delete “x1” in the denominator.

Line 118: All parameter should be in italic form.

Line 179: Differences in locations for both husk and cob were …

Line 182: Differences in genotypes and …

Line 184-: Please use abbreviations (MAC, MAC/e and so on) hereafter. Readers may feel confusing.

Line 210-216: This paragraph is too obvious for readers, and can be deleted.

Line 229: in a few locations

Line 245: Low but significant …  However, this statement is apparently inconsistent with that in lines 227-231. Which is correct? Please resolve this problem.

Line 248-: What are the “medium”, “low” and “high” groups? Readers find first these terms, because the results of cluster analysis was stated after this section. Revise the expression, or move and revise this paragraph after the cluster analysis.

Line 251: high group (A).

Line 266: I think that the check varieties should be included in the present cluster analysis, as an outer group. This may provide more valuable information.

Line 310-377: I think that the description of each trait are very tedious to read. Please describe essence and revise concisely, to avoid simple repeat of contents in tables, duplicate expression, and others.

Line 424: This statement is apparently inconsistent with the Line 421, “the correlations ….. in corn husk were low”. Revise it.

Line 428-433: Where are the correlation coefficients for these association? These data were missing. Did you determine the data of ears? Did you determine only the data of cob and husk, and not kernel? Readers cannot understand this paragraph.

Line 443: I believe this conclusion is completely a simple repetition of the previous section, involved no novel information, and should be deleted. If the authors want to state conclusion, please describe additional and valuable information in the conclusion than results and discussion.

Author Response

Dear Reviewer,

The manuscript entitled "Genotypic variation in anthocyanins, phenolic compounds and antioxidant activity in cob and husk of purple field corn" is revised according to the comments and suggestions of the reviewers. The revisions were highlighted by red letters and the track change of the revisions is also indicated.

The authors try to make the minimum change in the manuscript to avoid the possible errors.

The authors also very much appreciate the comments and suggestions of the reviewers to improve the manuscript.

Yours Sincerely,

Bhalang

Reviewer 2 Report

This is an interesting manuscript reporting the qualitative performance of purple corn and the GxE interaction for the traits studies.

Although the methodology is accurate there are various issues to be taken into account in the description and representation of results

1) The authors have completely overlooked the significance of the variation due to Location and to the GxE interaction.

As stated at L212-214, "...low variation across location implies that the traits are stable across different environments...". The concept cannot be applied to all traits. For some, the variation due to L is not trascurable in particular for those related to Husk (i.e MAC, TPC, DPPH, C). Moreover, all traits has a LxG interaction contributing for more than 10% with peaks of about 50%

Finally both L and LxG have evidenced a high significance (P<0.01)

Following the considerations above mentioned, did the author check which is the ranking of the genotypes in the two locations. As example the accession evidencing the high values for the traits scored in the Location 1 has the same trend in Location 2? All the genotypes studied are in the same groups (A-F) if we consider the locations separately? These information must be given.

Therefore, I would advice give the differencesof the means for the accessions in the two separate trials and then make cluster analysis indicating the "robust" clusters.

2) The description of results need to be improved, as example from L183 to L209 is just a description of the table which makes reading boring, i would suggest to highlight the most interesting results (i.e. Traits having the highest and lowest variation) referring to the table for the rest. 

This need to be applied to the whole description of the results. 

3) There is not much discussion. The discussion can be improved following the observations above indicated (1) and searching, if available, similar studies in non-purple corns

4) L248-L255 this part should go in the paragraph  3.2 in my opinion

5) Table 2 Chroma and Hue in general are traits highly correlated. Relatively to Husk, how is possible to explain so large variation between Locations for chroma? A separate analysis of the two trials could help to elucidate this

I would advice a major revision of the manuscript, a more interesting insight of the work will come out

Author Response

(The authors gave the same response as above.)

Round 2

Reviewer 1 Report

The revised manuscript (agronomy-381979-r1) was well improved. I believe this can be accepted for publication in Agronomy. Very tiny minor collections are as follows. It need not re-reviewing.

Line 90: … brown cob, and it is …  (?)

Line 91: … purple cob, and it is …(?)

Line 95: … white cob, and there are …(?)

Line 99: Delete “at” after “ two locations”.

Line 127: “1” should not be in italic.

Line 142: 2.5 mL water

Lines 177 and 178: top, middle and bottom

Author Response

Dear Reviewer, 

The manuscript entitled " Genotypic variation in anthocyanins, phenolic compounds and antioxidant activity in cob and husk of purple field corn" was revised in the second round according to the comments and suggestions of the reviewers. 

The authors respond to the reviewers point by point. 

The authors also check for possible errors, and the errors were revised and indicated by track change. 

Yours Sincerely, 

Bhalang

Reviewer 2 Report

The manuscript has been improved according to revision. I strongly encourage the authors to include photos of the genetic material studied even in a TOC figure, it would give a better view of the article.

Also it seems that conclusions have been deleted, i would not discard at all, but include few sentences

Author Response

(The authors gave the same response as above.)
